# A snapshot of mid Eocene landscapes in the southern Central Andes: Spore-pollen records from the Casa Grande Formation (Jujuy, Argentina)

Mariano J. Tapia[1☯], Ezequiel E. Farrell[2☯], Lilia R. Mautino[2], Cecilia del Papa[3], Viviana D. Barreda[1], Luis Palazzesi[1]*

1 Museo Argentino de Ciencias Naturales, Buenos Aires, Argentina (MACN-CONICET), 2 Centro de Ecología Aplicada del Litoral, Corrientes, Argentina (CECOAL-CONICET), 3 Centro de Investigaciones en Ciencias de la Tierra, Córdoba, Argentina (CICTERRA-CONICET)

☯ These authors contributed equally to this work.
* lpalazzesi@macn.gov.ar

**Data Availability Statement:** Data are available within the manuscript and its Supporting

## Abstract

The southern Central Andes–or Puna–now contains specialized plant communities adapted to life in extreme environments. During the middle Eocene (~40 Ma), the Cordillera at these latitudes was barely uplifted and global climates were much warmer than today. No fossil plant remains have been discovered so far from this age in the Puna region to attest to past scenarios. Yet, we assume that the vegetation cover must have been very different from what it looks today. To test this hypothesis, we study a spore-pollen record from the mid Eocene Casa Grande Formation (Jujuy, northwestern Argentina). Although sampling is preliminary, we found ~70 morphotypes of spores, pollen grains and other palynomorphs, many of which were produced by taxa with tropical or subtropical modern distributions (e.g., Arecaceae, Ulmaceae *Phyllostylon*, Malvaceae Bombacoideae). Our reconstructed scenario implies the existence of a vegetated pond surrounded by trees, vines, and palms. We also report the northernmost records of a few unequivocal Gondwanan taxa (e.g., *Nothofagus*, *Microcachrys*), about 5,000 km north from their Patagonian-Antarctic hotspot. With few exceptions, the discovered taxa–both Neotropical and Gondwanan–became extinct from the region following the severe effects of the Andean uplift and the climate deterioration during the Neogene. We found no evidence for enhanced aridity nor cool conditions in the southern Central Andes at mid Eocene times. Instead, the overall assemblage represents a frost-free and humid to seasonally-dry ecosystem that prevailed near a lacustrine environment, in agreement with previous paleoenvironmental studies. Our reconstruction adds a further biotic component to the previously reported record of mammals.

Information files. The fossil specimens analyzed and illustrated in the present study are housed in the public repository of the University of Jujuy (Argentina), Palynological Laboratory, under the code PAL JUA (P) Cz/Plg. N° 001a-g(Eo).

**Funding:** The authors received no specific funding for this work.

**Competing interests:** The authors have declared that no competing interests exist.

## Introduction

The Puna-Altiplano encompasses a high elevation montane grassland in the Central Andes, extending from southern Peru, through Bolivia into northern Chile and Argentina [1, 2]. The landscape is a high plateau ranging from 3,700 m to 4,500 m above sea level, with snow-covered peaks and saline lakes. The climate ranges from temperate to cold and arid; average temperature ranges between 0°C and 15°C while precipitation ranges between 200 and 300 mm per year [3]. These climatic conditions associated with important disturbances like droughts and frosts have restricted the development of life but have favored the establishment of species able to cope with these conditions. For example, the vegetation is characterized by tall tussocks of bunchgrass, and shrubs such as *Parastrephia lepydophilla* (Asteraceae) as well as other cushion plants and sparse lichens and mosses [4]. Fauna, on the other hand, include vicuña (*Lama vicugna*), guanaco (*Lama guanicoe*), chinchilla (*Chinchilla chinchilla*), vizcacha (*Lagidium viscacia*), the Andean cat (*Felis jacobita*) and the large flightless bird Darwin's rhea (*Rhea pennata*), among others [5–7].

The harsh climatic conditions prevailing in the Puna are mostly attributed to the Neogene uplift of the Andean Cordillera, which blocks the easterly humid winds, casting a shadow of dryness on the leeward [8–10]. However, the reconstructed climates from the Paleogene are remarkably different from those of today; during the Paleocene and early Eocene, sedimentological information (e.g., clay minerals and paleosols) from the Salta Group, in Northwestern Argentina, indicate humid-subtropical to tropical climates, although episodes of seasonality have been identified [11–14]. This paleoclimate supported a diverse biota with fish, crocodiles, lizards and mammals [15]. Also, spores and pollen grains demonstrated the existence of landscapes dominated by humid forests and lacustrine environments, with local arid conditions [16–21]. The onset of the Andean Cordillera uplift in the mid Eocene at these latitudes (22°S–26°S) [22, 23] have driven important environmental (e.g., [24]), and climatic shifts [25], although little is known about its impact on biotas (e.g., [26]). The available sedimentological information points to the existence of a depositional environment characterized by plains with localized ponds or shallow lakes and meandering fluvial systems with clay-rich floodplains [27]. Large herbivores as browsers (Leontiniidae) and some scavengers (Dasypodidae) along with small grazers, fruit and insect feeders (Prepidolopidae) have occurred across these landscapes [15, 28, 29]. Yet, little is known about the composition and structure of the plant cover, which is the primary support of habitats where these and other herbivores may have lived, fed on, and found shelter. Here, we present terrestrially-derived palynomorphs (largely pollen and spores) from deposits of the Casa Grande Formation (Figs 1 and 2), middle Eocene (approx. 40 Myr), exposed in northwestern Argentina. The Casa Grande Formation largely includes reddish oxidized sediments, where palynomorphs are substantially less abundant. Yet we were able to recover a productive layer with microscopic remains produced by a wide range of organisms (i.e., plants, fungi, insects).

### Geological and chronostratigraphic context

The Casa Grande Formation [30] includes red to reddish brown siltstones and sandstones (Fig 2) with a mean thickness of 800 m [31], and crops out along the Tres Cruces–Mina Aguilar area, in the Jujuy province, northwestern Argentina (Fig 1). This unit represents the onset deposition in the foreland basin [27, 31] contemporaneous with the initial construction of the Andean topography [23]. Montero-López et al. [27] proposed a subdivision of this sedimentary unit into the Casa Grande 1 and Casa Grande 2 sequences on the basis of a new and more detailed mapping analysis (Fig 3). The Casa Grande 1 sequence was previously assigned to the Lumbrera Formation of the Salta Group (e.g., [31, 32]). However, Montero-López et al. [27]

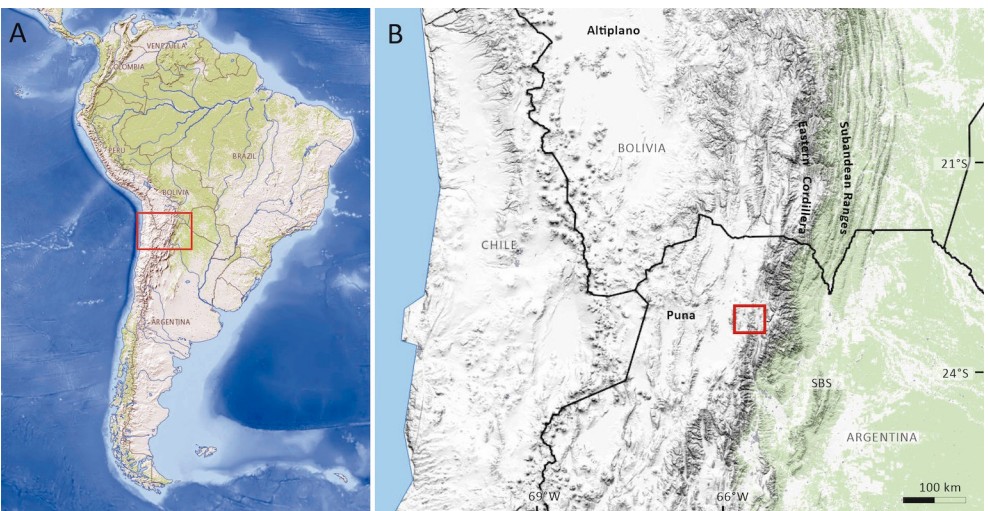

**Fig 1. Location map.** (A) Map of South America indicating the region covered by the present study. (B) Map of northern Chile, southern Bolivia and northwestern Argentina, indicating the sample site (Casa Grande Fm, Jujuy Province). Maps were taken from USGS National Map Viewer (public domain).

recognized a sharp stratigraphic contact between the underlying dark reddish brown, highly cemented mudstones of the early Eocene Lumbrera Formation and the overlying reddish, gypsum-rich sandy siltstones of the Casa Grande 1 sequence. The ~160 m- thick Casa Grande 1 sequence, which preserves the palynomorphs we present and illustrate in this contribution,

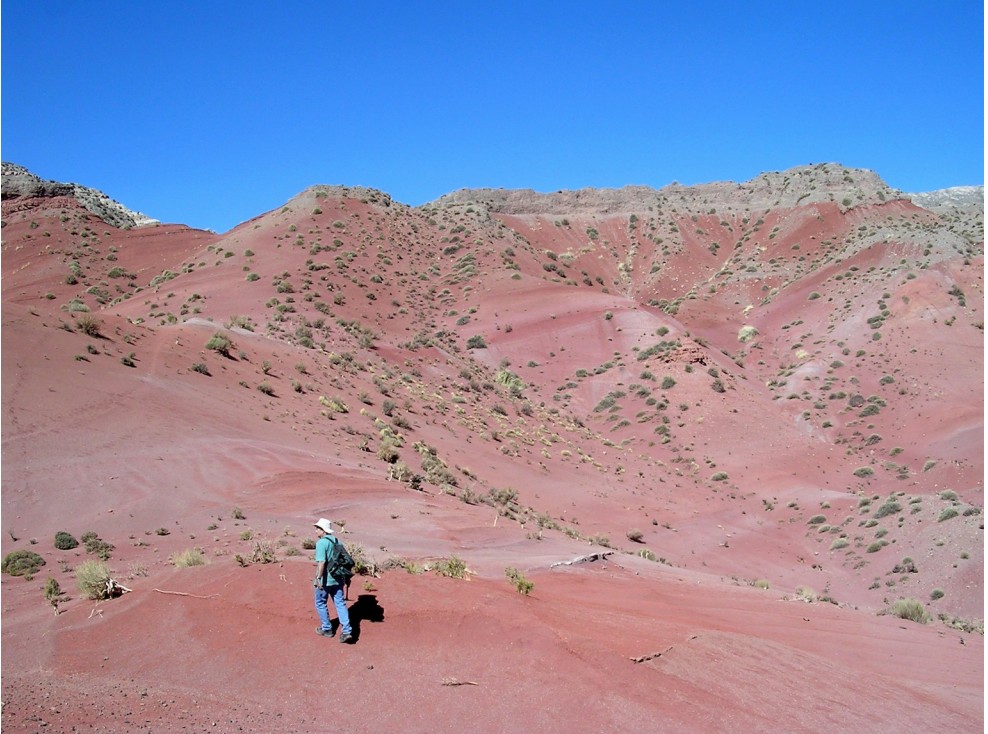

**Fig 2. Field aspect of the palynomorph-bearing sediments.** Note the overall appearance of the terrestrial deposits cropping out in Jujuy (Casa Grande Fm.).

| Late Miocene-Pliocene<br><br>Early Miocene<br><br>Middle Eocene-Oligocene? | Undifferentiated Groups. | | Pisungo Fm.<br><br>Río Grande Fm.<br><br>Casa Grande II Fm.<br><br>Casa Grande I Fm. |
|---|---|---|---|
| Lower Cretaceous-early Eocene | Salta Gr. | Santa Bárbara Sbgr. | Lumbrera Fm.<br><br>Maíz Gordo Fm.<br><br>Mealla Fm. |
| | | Balbuena Sbgr. | Tunal/Olmedo Fm.<br>Yacoraite Fm.<br>Lecho Fm. |
| | | Pirgua Sbgr. | Undifferentiated units. |
| Ordovician | Santa Victoria Gr. | | |
| Cambrian | Mesón Gr. | | |
| Precambrian-Cambrian | Puncoviscana Fm. | | |

**Fig 3. Stratigraphic chart of the Groups/Formations recognized in the region.** Note the stratigraphic relationship between the Casa Grande I Fm. and other units used for comparison in the present analysis (Tunal Fm./Olmedo Fm., Mealla Fm., Maíz Gordo Fm., and Lumbrera Fm.).

includes dark reddish orange fine-grained sediments, interbedded with green to dark gray claystones, carbonate and fine-grained sandstones (Fig 4). The depositional environment for this sequence is interpreted to be transitional between a vegetated mudflat with ponds and shallow lakes, and isolated channel belts. Until now, no radiometric ages have been published for the Casa Grande 1 sequence, although absolute age for the upper Lumbrera Formation–equivalent unit–provided an absolute U/Pb zircon depositional age of 39.9 ± 0.4 Ma (Eocene) [33]. Furthermore, fossil mammalian remains discovered from the Casa Grande 1 also support the Eocene age (e.g., [28, 34]).

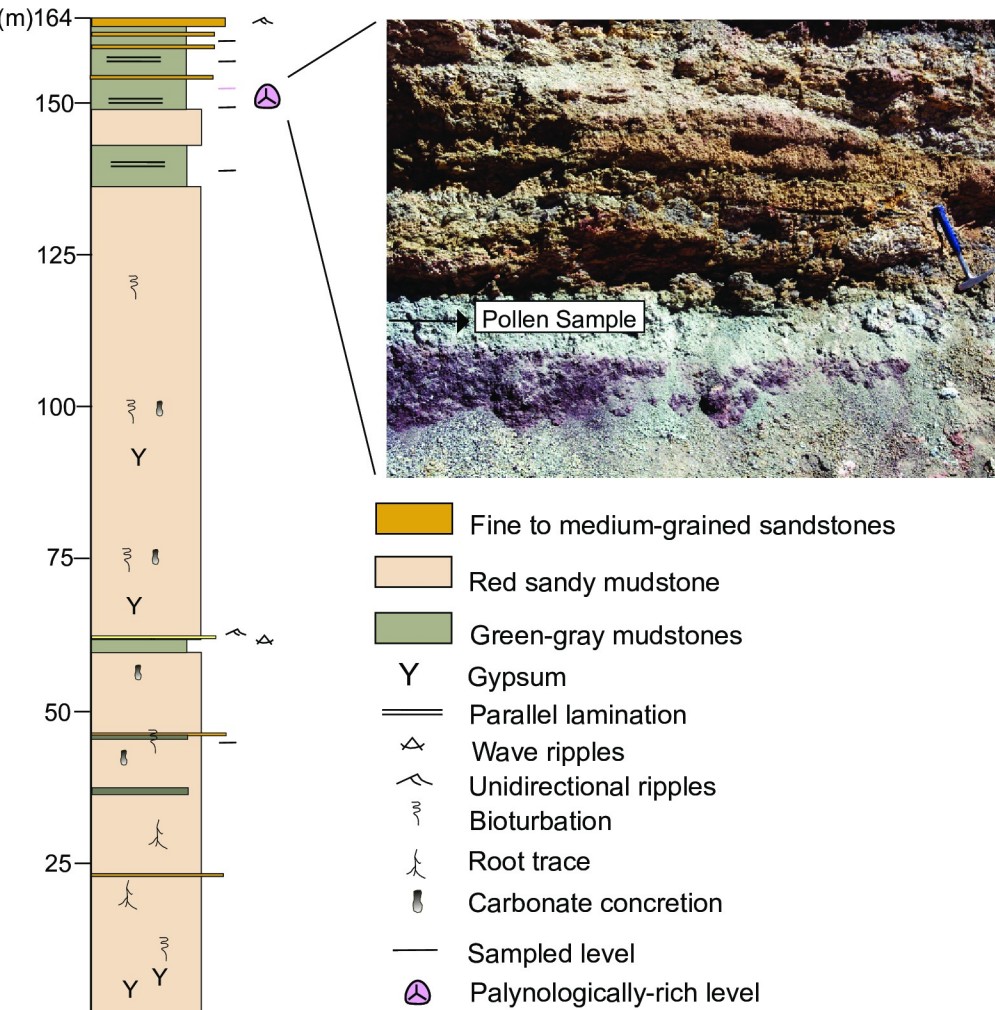

**Fig 4. Stratigraphic and sedimentological characteristics of the Casa Grande Formation.** Note the location of the palynologically-rich level located in the upper part of the section.

## Material and methods

Palynological samples were collected from seven stratigraphic levels of the Casa Grande 1 sequence from a 165 m-thick section of red to reddish gypsum-rich sandy siltstones and green-gray mudstones, exposed in the Casa Grande valley (Figs 1–4). Permits to access and collect sedimentary samples were given by indigenous communities from the region (i.e., Casa Grande, Vizcarra and El Portillo). Palynomorphs are best preserved in unoxidized, fine-grained, primarily dark-colored (gray to black) sediments; the Casa Grande 1 sequence lacks suitable lithologies to preserve organic material, and hence any productive palynological level is very valuable for reconstructing past floras and environments. Seven samples were processed and sieved using standard palynological techniques [35, 36]), of which only one, from a lacustrine bed, yielded palynomorphs; the residue was permanently mounted in six palynological replicates using UV-curable acrylates gel [37]. The slides are housed in the palynological collection of Jujuy under the code PAL JUA (P) Cz/Plg. N° 001a-g(Eo). Photomicrographs were taken with a Leica ICC50W camera; the coordinates of the illustrated specimens refer to a Leica DM500 microscope. Pollen terminology follows Punt et al. [38]. Fossil spores and pollen

grains were identified to species level and diagnosed to modern families, subfamilies, tribes or genera whenever possible (Table 1). We counted 546 palynomorphs (S1 Table). We used R software program, version 2.2.0 [39] for estimating similarities among the spore-pollen assemblages from the Casa Grande 1 sequence and those previously reported from other Paleogene localities cropping out in northwestern Argentina [i.e., Tunal/Olmedo Formation (early Paleocene [19–21]); Mealla Formation (late Paleocene [17]); Maíz Gordo Formation (Paleocene/Eocene [18]); Lumbrera Formation (Early Eocene [16])] based on presence/absence data (S2 Table) because most of the published works lack pollen counts (S3 Table). The Hierarchical Clustering ('hclust' function) was run using a distance matrix ('dist' function, 'method: binary) from the R package 'stats' version 4.0.2. The number of samples and spore-pollen species recovered for each of the sedimentary sections used for comparisons vary ([16–21], S3 Table) and hence results should be interpreted accordingly: Tunal Fm (14 samples, 42 palynomorphs); Olmedo Formation (~11samples, 32 palynomorphs); Mealla Formation (4 samples, 36 palynomorphs); Maíz Gordo Formation (4 samples, 37 palynomorphs); Lumbrera Formation (12 samples, 19 palynomorphs).

## Results and discussion

The palynological assemblage recovered from the Casa Grande 1 sequence is relatively well preserved and relatively diverse. It includes spores, pollen grains, algae, among other fossil remains (fungi and zoological remains). We identified 43 spore and pollen species (Table 1, Figs 5–7), including 2 bryophytes, 4 pteridophytes, 3 gymnosperms, and 34 angiosperms. From these, we found the nearest living relatives (families, subfamilies, tribes or genera) of 37 species (2 ferns, 2 bryophytes, 3 gymnosperms, and 30 angiosperms); the remaining sporomorphs have doubtful or uncertain botanical affinity. From the 43 recognized taxa, 38 has been reported for the first time at Casa Grande 1 Fm., compared to the older Paleogene sediments cropping out nearby (i.e., Olmedo, Tunal, Mealla, Maíz Gordo, Lumbrera). These records, combined with those recorded earlier in northwestern Argentina, provide a total of 55 confidently assigned plant families for most of the early Paleogene (Paleocene-mid Eocene).

The fossil palynomorphs preserved in the terrestrial sediments of the Casa Grande 1 Formation come from a single sedimentary layer, yet they were transported from local to more distant sources by biotic (e.g., insects) or abiotic (e.g., wind, rivers) means. For example, we estimate that some of the non-pollen palynomorphs were preserved *in situ* or very close to the fossil site because they exhibit few adaptations to travel long distances. This is the case of palynomorphs assigned to algae such as Zygnemataceae (*Mougeotia* sp., Fig 7N; *Spirogyra* sp. 6, Fig 7Q; *Stigmozygodites ministigmosus*, Fig 7P), Scenedesmaceae (*Scenedesmus* sp., Fig 7R) and Hydrodictyaceae (*Pediastrum biradiatum*; *P. boryanum*, Fig 7M and *P. duplex* var. *gracillimum*), which strictly occur in freshwater ecosystems, as well the presence of fungal spores of *Reduviasporonites* suggests shallow aquatic habitats [40]. In addition, Filinia-type resting eggs (Fig 7K) are common in fresh-brackish ponds and lakes [41, 42]. Moreover, oribatid mites as *Hydrozetes* remains (Fig 7J) could also be restricted to freshwater habitats too (i.e., pools, ponds, raised bogs, lakes, slowly flowing waters, and submerse vegetation) [43]. Long distance transport of these palynomorphs (as coenobia or zygospores) has rarely been reported (e.g., [44, 45]). Additionally, the several fossil fungal spores recovered (e.g., *Biporipsilonites krempii*, Fig 7A; *Inapertisporites edigeri*, Fig 7B; *Pluricellaesporites sheffyi* (not illustrated); *Multicellites crassisporus*, Fig 7C; *Scolecosporites modicus*, Fig 7F; *Polycellaesporonites bellus*, Fig 7E) may have been produced at a short distance from the place where sporulation took place [41], hence indicating local occurrence. The presence of these sporomorphs at the fossil site, particularly those assigned to algae, indicate the occurrence of a shallow lake, or a pond, in

**Table 1. Identified morphotype list.**

| Fossil taxon | Botanical affinity |
|---|---|
| **Triletes pores** | |
| *Deltoidospora minor* (Couper) Pocock 1970 (Fig 5B) | Cyatheaceae, Dicksoniaceae, Schizaeaceae |
| *Leiotriletes* sp. Mautino 2010 (Fig 5C) | Pteridophyta |
| *Polypodiaceoisporites* cf. P. *retirugatus* Muller 1968 (Fig 5D) | Pteridaceae (*Pteris*) |
| *Reboulisporites fuegiensis* Zamaloa and Romero 1990 (Fig 5E) | Ricciaceae |
| *Todisporites minor* Couper 1958 (Fig 5A) | Osmundaceae |
| *Zlivisporis* sp. (Fig 5F) | Bryophyta, Hepaticae |
| **Sacatepollen** | |
| *Microcachryidites antarcticus* Cookson 1947 (Fig 5H) | Podocarpaceae (*Microcachrys tetragona*) |
| *Podocarpidites* sp. (Fig 5G) | Podocarpaceae |
| **Plicatepollen** | |
| Equisetosporites notensis (Cookson) Romero 1977 (Fig 5I) | Ephedraceae |
| **Inaperturatepollen** | |
| *Smilacipites* sp. cf. S. *herbaceoides* Wodehouse 1933 (Fig 5J) | Smilacaceae (*Smilax*, *Peltandra*) |
| **Monosulcatepollen** | |
| *Arecipites minutiscabratus* (McIntyre 1968) Milne 1988 (Fig 5K) | Arecaceae |
| *Echimonocolpite s*sp. (Fig 5N) | Arecaceae/unknown angiosperm? |
| *Liliaciditesmirus* Srivastava 1969 (Fig 5M) | Liliaceae |
| *Liliacidites vermireticulatus* Archangelsky and Zamaloa 1986 (Fig 5L) | Iridaceae |
| *Punctilongisulcites punctiechinatus* (Krutzsch) Casas Gallego and Barrón 2020 | Hydrocharitaceae |
| *Verrumonocolpites* sp. | Arecaceae |
| **Colpate pollen** | |
| *Beaupreadites* sp. (Fig 6A) | Proteaceae (*Beauprea*) |
| *Nothofagidites anisoechinatus* Menendez and Caccavari 1965 (Fig 5O) | Nothofagaceae Brassi Type |
| *Nothofagidites saraensis* Menendez and Caccavari 1975 (Fig 5P) | Nothofagaceae Fusca Type |
| *Retitrescolpitesadultus* González Guzmán 1967 (Fig 5Q) | Unknown angiosperm |
| *Tricolpites aspermarginis* McIntyre 1968 (Fig 5R) | Violaceae |
| *Tricolpites membranus* Couper 1960 (Fig 5S) | Phytollacaceae (*Seguieria*) |
| *Tricolpites* sp. Mautino 2009 (Fig 5T) | Oxalidaceae (*Oxalis*) |
| **Colporate pollen** | |
| *Ailanthipites* sp. (Fig 6E) | Anacardiaceae |
| *Baumannipollis* sp. (Fig 6J) | Malvaceae (*Lagunaria*, *Modiolastrum*, *Urocarpidinium*, *Tarasa*) |
| *Bombacacidites* sp. (Fig 6F) | Malvaceae Bombacoideae |
| *Foveotricolporites* sp. (Fig 6C) | Unknown angiosperm |
| *Heterocolpites rotundus* Hoorn 1993 (Fig 6I) | Combretaceae-Melastomataceae |
| *Malvacipolloides tucumanensis* Mautino et al. 2004 | Malvaceae, Malveae |
| *Margocolporites tenuireticulatus* Barreda 1997 (Fig 6D) | Fabaceae Mimosoideae |
| *Rhoipites baculatus* Archangelsky 1973 (Fig 6G) | Fabaceae Papilionoideae (*Aeschynomene*) |
| *Rhoipites guianensis* (Van der Hammen and Wymstra) Jaramillo and Dilcher 2001 (Fig 6H) | Malvaceae Sterculioideae (*Firmiana* and *Hildegardia*) |
| *Siltaria dilcheri* Silva-Caminha et al. 2010 (Fig 6B) | Unknown angiosperm |
| **Porate pollen** | |
| *Corsinipollenites menendezii* Quattrocchio 1978 (Fig 6M) | Onagraceae (*Ludwigia*) |
| *Gomphrenipollis* sp. 1 (Fig 6S) | Amaranthaceae (*Gomphrena*) |
| *Gomphrenipollis* sp. 2 (Fig 6T) | Amaranthaceae (*Gomphrena*) |

(*Continued*)

**Table 1.** (Continued)

| Fossil taxon | Botanical affinity |
|---|---|
| *Graminidites* sp. (Fig 6K) | Poaceae |
| *Pandaniidites* sp. (Fig 6L) | Araceae (*Lemna*) |
| *Periporopollenites polyoratus* (Couper 1960) Stover in Stover and Partridge 1973 (Fig 6R) | Caryophyllaceae-Trimeniaceae |
| *Periporopollenites* sp. (Fig 6Q) | Caryophyllaceae (*Silene*) |
| *Psilaperiporites circinatus* D'Apolito et al. 2021 (Fig 6P) | Unknown angiosperm |
| *Psilatriporites desilvae* Hoorn 1993 (Fig 6N) | Fabaceae Caesalpinioideae |
| *Verrustephanosporites simplex* Leidermeyer 1966 (Fig 6O) | Ulmaceae (*Phyllostylon*) |
| **Fungal remains** | |
| *Biporipsilonites krempii* (Varma and Rawat 1963) Kalgutkar and Jansonius 2000 (Fig 7A) | Fungi |
| *Colligerites kutchensis* (Kar and Saxena) Jain and Kar 1979 | Fungi |
| *Dictyosporites* sp. Kalgutkar and Braman 2008 | Fungi |
| *Inapertisporites edigeri* Kalgutkar and Jansonius 2000 (Fig 7B) | Fungi |
| *Inapertisporites subovoideus* (Sheffy and Dilcher) Kalgutkar and Jasonius 2000 | Fungi |
| *Multicellites crassisporus* (Salard-Cheboldaeff and Locquin) Kalgutkar and Jansonius 2000 (Fig 7C) | Fungi |
| *Pluricellaesporites sheffyi* Martínez Hernández and Tomasini Ortiz 1989 | Fungi |
| *Polycellaesporonites bellus* Chandra et al. 1984 (Fig 7E) | Fungi |
| *Quilonia* cf. *Q. allepeyensis* (Ramanujam and Rao) Kalgutkar and Jasonius 2000 (Fig 7D) | Fungi |
| *Reduviasporonites* cf. *R. catenulatus* Wilson 1962 | Fungi |
| *Scolecosporites modicus* Kalgutkar and Jansonius 2000 (Fig 7F) | Fungi |
| **Zooclasts** | |
| *Filinia*-type resting eggs (Fig 7K) | Rotifera, Trochosphaeridae |
| *Hydrozetes* adult leg (Fig 7J) | Acari, Oribatida |
| Scales of Lepidopteran wings (Fig 7G and 7H) | Lepidoptera |
| Scolecodont-like palynomorphs (Fig 7I) | Arthropoda? |
| **Other terrestrial palynomorphs** | |
| Compact type sclerocyte (Fig 7L) | - |
| **Fresh-water algae** | |
| *Cymatiosphaera* sp. Mays et al. 2021 | Cymatiosphaeraceae |
| *Mougeotia* sp. (Fig 7N) | Zygnemataceae |
| *Pediastrum biradiatum* Meyen 1829 | Hydrodictyaceae |
| *Pediastrum boryanum* (Turp.) Menegh 1840 (Fig 7M) | Hydrodictyaceae |
| *Pediastrum duplex* var. *gracillimum* West and West 1895 | Hydrodictyaceae |
| *Planctonites stellarius* (Potonié 1934) Krutzsch 1960 (Fig 7O) | Zygnematales, Desmidiaceae |
| *Scenedesmus* sp. (Fig 6R) | Scenedesmaceae |
| *Spirogyra* sp. 6 Martinez et al. 2008 (Fig 7Q) | Zygnemataceae |
| *Stigmozygodites ministigmosus* Krutzsch and Pacltová 1990 (Fig 7P) | Zygnemataceae (*Zygnema*) |
| **Marine algae** | |
| Indeterminate Acritarch cyst | Unknwon |
| Indeterminate Dinoflagellate cyst 1 (Fig 7S) | Dinoflagellata |
| Indeterminate Dinoflagellate cyst 2 (Fig 7T) | Dinoflagellata |

Palynomorphs recovered from the Casa Grande Formation along with their botanical affinities (Nearest Living Relative).

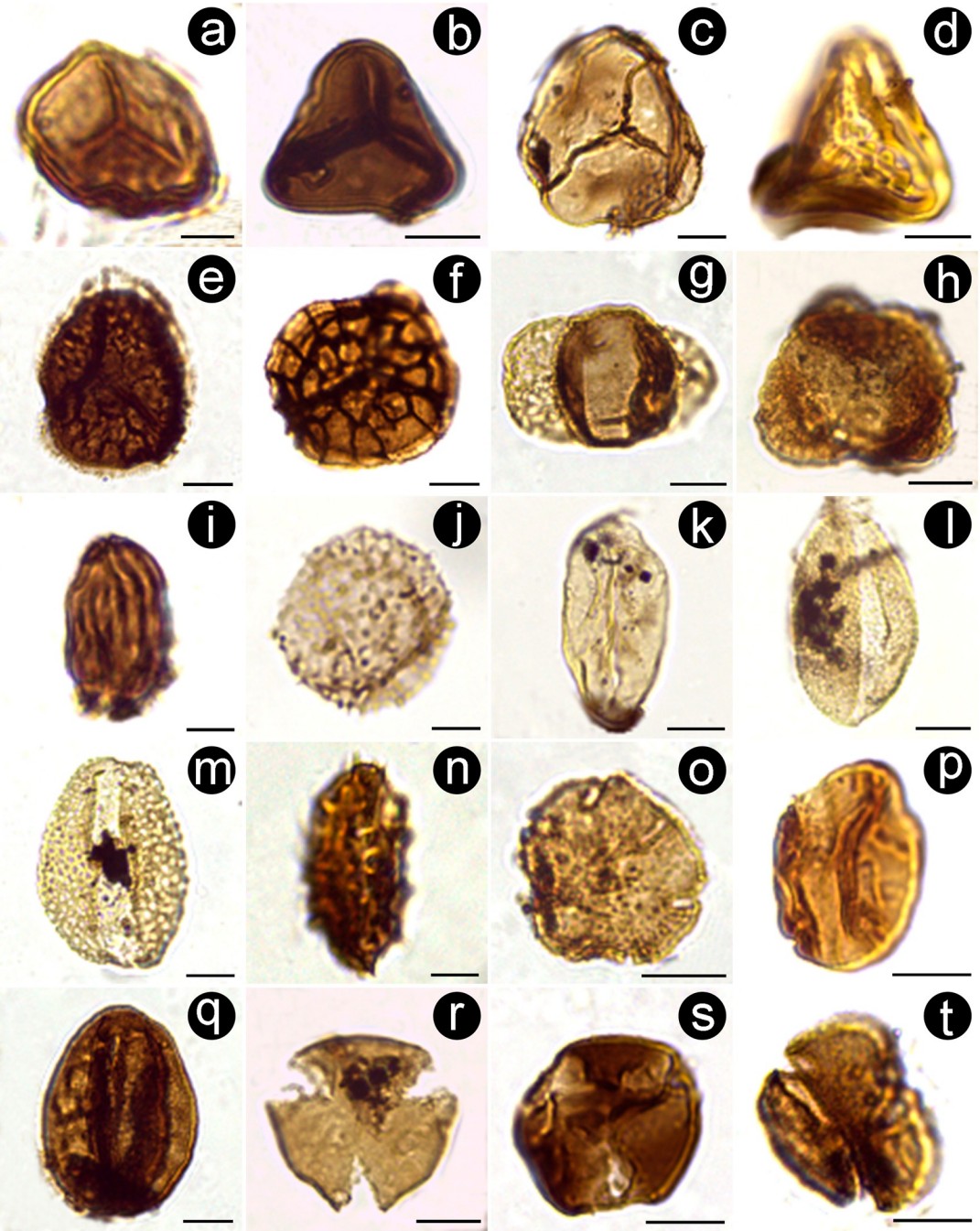

**Fig 5. Microscope images of selected spores and pollen grains from the Casa Grande Formation.** (A) *Todisporites minor* (PAL JUA (P) Cz/Plg. N° 001c(Eo)+10 µm: 21.9–150.9). (B) *Deltoidospora minor* (PAL JUA (P) Cz/Plg. N° 001d(Eo)+10 µm: 10–135). (C) *Leiotriletes* sp. (PAL JUA (P) Cz/Plg. N° 001c(Eo)+10 µm: 9–138.9). (D) *Polypodiaceoisporites* cf. *P. retirugatus* (PAL JUA (P) Cz/Plg. N° 001b(Eo)+10 µm: 21.5–131). (E) *Reboulisporites fuegiensis* (PAL JUA (P) Cz/Plg. N° 001e(Eo)+25 µm: 3.5–139.4). (F) *Zlivisporis* sp. (PAL JUA (P) Cz/Plg. N° 001f(Eo)+25 µm: 16–151). (G) *Podocarpidites* sp. (PAL JUA (P) Cz/Plg. N° 001b(Eo)+10 µm: 3–153.5). (H) *Microcachryidites antarcticus* (PAL JUA (P) Cz/Plg. N° 001d(Eo)+10 µm: 3.9–130.5). (I) *Equisetosporites notensis* (PAL JUA (P) Cz/Plg. N° 001e(Eo)+10 µm: 4.5–154.6). (J) *Smilacipites* cf. *S. herbaceoides* (PAL JUA (P) Cz/Plg. N° 001d(Eo)+10 µm: 16.9–135.5). (K) *Arecipites minutiscabratus* (PAL JUA (P) Cz/Plg. N° 001d(Eo)+10 µm: 23–152.2). (L) *Liliacidites vermireticulatus* (PAL JUA (P) Cz/Plg. N° 001d(Eo)+10 µm: 24.2–126.9). (M) *Liliacidites mirus* (PAL JUA (P) Cz/Plg. N° 001d(Eo)+25 µm: 15–139). (N) *Echimonocolpites* sp. (PAL JUA (P) Cz/Plg. N° 001b(Eo)+10 µm: 5.2–128.9). (O) *Nothofagidites anisoechinatus* (PAL JUA (P) Cz/Plg. N° 001e(Eo)+10 µm: 13–128). (P) *Nothofagidites saraensis* (PAL JUA (P) Cz/Plg. N° 001e(Eo)+10 µm: 15.5–152.6). (Q) *Retitrescolpites adultus* (PAL JUA (P) Cz/Plg. N° 001d(Eo)+25 µm: 14–150.4). (R)*Tricolpites asperamarginis* (PAL JUA (P) Cz/Plg. N° 001d(Eo)+10 µm: 13.8–128.8). (S) *Tricolpites membranus*

(PAL JUA (P) Cz/Plg. N° 001f(Eo)+10 µm: 14.9–152). (T) *Tricolpites* sp. (PAL JUA (P) Cz/Plg. N° 001c(Eo)+10 µm: 17–143. Scale bars: 10 µm (except in A, I, J, and N: 5 µm). Each morphotype name is followed by slide number and coordinates of Leica DM500 microscope held at the Museo Argentino de Ciencias Naturales, Buenos Aires, Argentina.

agreement with previous sedimentological studies at the Casa Grande Formation [27]. Furthermore, other microscopic remains such as wing scales indicate the presence of moths or butterflies in the surroundings. Peaks in abundance of these fragments can be used by some authors to reconstruct insect outbreaks in ancient forests [46, 47]. Apart from that, scolecodont-like palynomorphs (Fig 7I) could be associated with mandibles or even body parts of a variety of freshwater arthropods (adults and larval stages) such as ants, copepods, oribatid mites, among others [48–50].

Other palynomorphs, such as spores of bryophytes and pteridophytes, may have also been produced by parent plants growing nearby. This is the case of bryophytes (*Reboulisporites fuegiensis*, Fig 5E; *Zlivisporis* sp., Fig 5F) and ferns [e.g., (Cyatheaceae/Dicksoniaceae, *Deltoidospora minor*, Fig 5B); (Osmundaceae; *Todisporites minor*, Fig 5A); (Pteridaceae, *Polypodiaceoisporites* cf. *P. retirugatus*, Fig 5D)]. We assume these spores have been transported by watercourses from relatively short distances. The presence of pollen grains assigned to aquatic plants such as water-plantains (Onagraceae, *Corsinipollenites menendezii*, Fig 6M) and duckweeds (Lemnoideae, Araceae; *Pandaniidites* sp., Fig 6L) also supports the occurrence of relatively quiet water bodies.

Other palynomorphs may have been produced within a short distance of the fossil site such as Osmundaceae (*Todisporites minor*, Fig 5A), Ricciaceae (*Rebulisporites fuegiensis*, Fig 5E), which most likely were transported to the pond from a more distant source either by insects or watercourses. Additionally, we suggest that the pollen grains of the Phytolaccaceae (*Tricolpites membranus*, Fig 5S), Fabaceae (*Rhoipites baculatus*, plate Fig 6G), Fabaceae Caesalpinioideae (*Margocolporites tenuireticulatus*, Fig 6D), the palm family Arecaceae (*Arecipites minutiscabratus*, Fig 5K; *Verrumonocolpites* sp.), Liliaceae (*Liliaciditesmirus*, Fig 5M), Ulmaceae *Phyllostylon* (*Verrustephanosporites simplex*, Fig 6O), Smilacaceae (*Smilacipites* cf. *S. herbaceoides*, Fig 5J), and Malvaceae Bombacoideae (*Bombacacidites* sp., Fig 6F) may have grown near the pond. Many of these (sub)families are well represented (S1 Table), including species with predominantly subtropical and seasonally dry modern distributions [51–53] and we suggest that they may have developed as patches near the pond, or as gallery forest along rivers (Fig 8). Other pollen species that became transported and preserved at the pond include the family Ephedraceae (*Equisetosporites notensis*, Fig 5I), commonly occurring throughout salt-stress and alkali-stress areas. Among non-pollen palynomorphs, fungi spores such as *Multicellites crassisporus*, Fig 7C; and *Pluricellaesporites sheffyi* (not illustrated) reinforce the hypothesis of humid conditions [40, 54].

Pollen grains with clear long-distance adaptations such as those produced by southern beeches (*Nothofagidites anisoechinatus* and *N. saraensis*, Fig 5O–5P) and podocarps (*Podocarpidites* sp. and *Microcachryidites antarcticus*, Fig 5G and 5H) may have been blown in, probably from altitudinally zoned or edaphically restricted regions. The presence of these Gondwanan elements in tropical regions must not be surprising as today some of them (e.g., *Podocarpus*) coexist in the Southern Andean Yungas; a very humid narrow band of forest between the drier Gran Chaco region to the east and the dry, high altitude Puna region to the west. Some of the Gondwanan taxa that became regionally extinct in the Puna still prevail southwards in the humid and cool Andean region of Patagonia, such as the case of the southern beech genus *Nothofagus*. Additionally, the Tasmanian genus *Microcachrys* was detected in the Puna which is represented by low shrubs growing up to 1 m tall at high altitudes. It

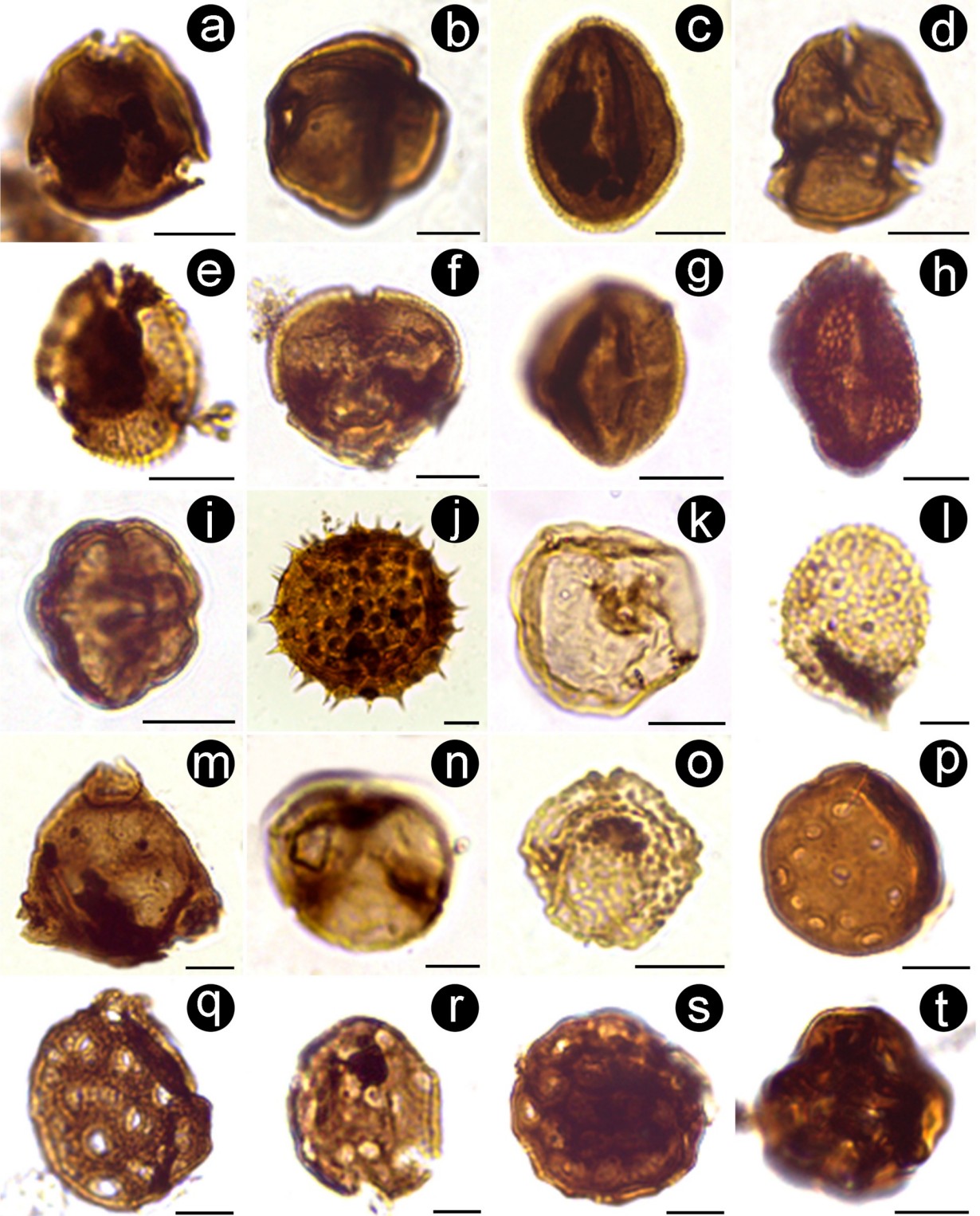

**Fig 6. Microscope images of selected pollen grains from the Casa Grande Formation.** (A) *Beaupreadites* sp. (PAL JUA (P) Cz/Plg. N° 001b (Eo)+10 μm: 7.5–142). (B) *Siltaria dilcheri* (PAL JUA (P) Cz/Plg. N° 001d(Eo)+10 μm: 17–151.2). (C) *Foveotricolporites* sp. (PAL JUA (P) Cz/Plg. N° 001b(Eo)+10 μm: 15.5–141.5). (D) *Margocolporites tenuireticulatus* (PAL JUA (P) Cz/Plg. N° 001a(Eo)+10 μm: 23–141). (E) *Ailanthipites* sp. (PAL JUA (P) Cz/Plg. N° 001b(Eo)+10 μm: 19–149). (F) *Bombacacidites* sp. (PAL JUA (P) Cz/Plg. N° 001e(Eo)+10 μm: 8–153.2). (G) *Rhoipites baculatus* (PAL JUA (P) Cz/Plg. N° 001c(Eo)+10 μm: 24–149.9). (H) *Rhoipites guianensis* (PAL JUA (P) Cz/Plg. N° 001e(Eo)+10 μm: 10–152). (I)

*Heterocolpites rotundus* (PAL JUA (P) Cz/Plg. N˚ 001e(Eo)+10 μm: 9.5–152.5). (J) *Baumannipollis* sp. (PAL JUA (P) Cz/Plg. N˚ 001c(Eo)+25 μm: 21.3–138). (K) *Graminidites* sp. (PAL JUA (P) Cz/Plg. N˚ 001c(Eo)+10 μm: 15–145.9). (L) *Pandaniidites* sp. (PAL JUA (P) Cz/Plg. N˚ 001d(Eo) +10 μm: 13.9–130.2). (M) *Corsinipollenites menendezii* (PAL JUA (P) Cz/Plg. N˚ 001d(Eo)+25 μm: 2.5–137). (N) *Psilatriporites desilvae* (PAL JUA (P) Cz/Plg. N˚ 001d(Eo)+10 μm: 16.7–155). (O) *Verrustephanosporites simplex* (PAL JUA (P) Cz/Plg. N˚ 001d(Eo)+10 μm: 14,8–142,7). (P) *Psilaperiporites circinatus* (PAL JUA (P) Cz/Plg. N˚ 001f(Eo)+10 μm: 10.1–132.2). (Q) *Periporopollenites* sp. (PAL JUA (P) Cz/Plg. N˚ 001e(Eo) +10 μm: 7–151). (R) *Periporopollenites polyoratus* (PAL JUA (P) Cz/Plg. N˚ 001e(Eo)+10 μm: 9.5–152). (S) *Gomphrenipollis* sp. 1 (PAL JUA (P) Cz/Plg. N˚ 001e(Eo)+10 μm: 7–141). (T) *Gomphrenipollis* sp. 2 (PAL JUA (P) Cz/Plg. N˚ 001b(Eo)+10 μm: 4–130.5). Scale bars: 10 μm (except in L, N and R: 5 μm). Each morphotype name is followed by slide number and coordinates of Leica DM500 microscope held at the Museo Argentino de Ciencias Naturales, Buenos Aires, Argentina.

completely disappeared from South America, probably during the Miocene, after a major bloom of Gondwanan elements in Patagonia during the Oligocene [55].

A comparison with assemblages recovered from older units in northwestern Argentina (i.e., Tunal/Olmedo, Mealla, Maíz Gordo and Lumbrera formations) reveals that the assemblage from the Casa Grande 1 formation is quite different (Fig 9, S2 Table), particularly due to the reduced number of species in common. In fact, our cluster analysis (based on presence-absence data) shows a dramatically different hierarchical scheme with one major cluster including all Paleocene-early Eocene assemblages, all sister to the mid Eocene Casa Grande 1 Fm. as a single tip (Fig 9). Note that the stratigraphically closest Paleogene assemblages are clustered together, that is early Paleocene assemblages (Tunal/Olmedo Fm.) on one side and middle Paleocene (Mealla Fm.) and Paleocene/Eocene (Maíz Gordo Fm.) assemblages on the other. The slightly younger Lumbrera Formation stands next to those assemblages in our cluster analysis.

The differences between our palynological assemblage and those previously studied from the Salta Basin–also supported by our cluster analysis–can be either because; 1) The number of species identified in our study from the Casa Grande 1 Fm. is dramatically different from those previously reported from older deposits (taxonomic bias hypothesis); 2) The environmental conditions prevailing during the accumulation of the palynomorph bearing level of the Casa Grande 1 Fm. favored the preservation of more spores and pollen types (taphonomic bias hypothesis); 3) The uplift of the Andean range (mid Eocene) may have driven shifts in the paleo-floras and, by extension, in the palynological assemblages (paleoecological hypothesis); 4) The increasing concentrations of atmospheric $CO_2$ during the mid-Eocene climatic optimum probably triggered shifts in plant communities (paleoclimatic hypothesis); 5) The Casa Grande Fm. would be slightly younger than previously thought and therefore new and more modern floristic components were recorded (stratigraphical hypothesis). Among these, we reject hypothesis 1 because previous palynological works (e.g., [17–19, 56]) have been exhaustive, that is, they described and illustrated a large number of palynomorphs using a reasonable number of samples. In fact, the number of species identified for Casa Grande Fm. (43 species, this study) is very close to, for example, the 42 species identified for the Tunal Fm. (S2 Table). We tentatively reject hypothesis 2 because the reconstructed environmental conditions appear to have been terrestrial (i.e., ponds, swamps or shallow lakes) for all Paleogene sites from the region. We infer hypotheses 3 and 4 are the most likely largely because of the rise of typical arid-adapted elements (e.g., Poaceae and Ephedraceae), may have been driven, at least in part, by the effect of the initial stages of the Andean uplift. These families are common today in the southern Puna, and appear to have coexisted in the region since the accumulation of the Casa Grande Fm. (mid Eocene times) and onwards. On the other hand, the climatic optimum of the mid Eocene, promoted by an increasing concentration of atmospheric $CO_2$, may have favored the rise of tropical elements; for example, we identified several species of palms and other tropical-affinity lineages (e.g., Malvaceae Bombacoideae, Smilacaceae). Finally, we cannot entirely reject hypothesis 5 because the pollen-bearing sediments of the Casa Grande Fm. can be

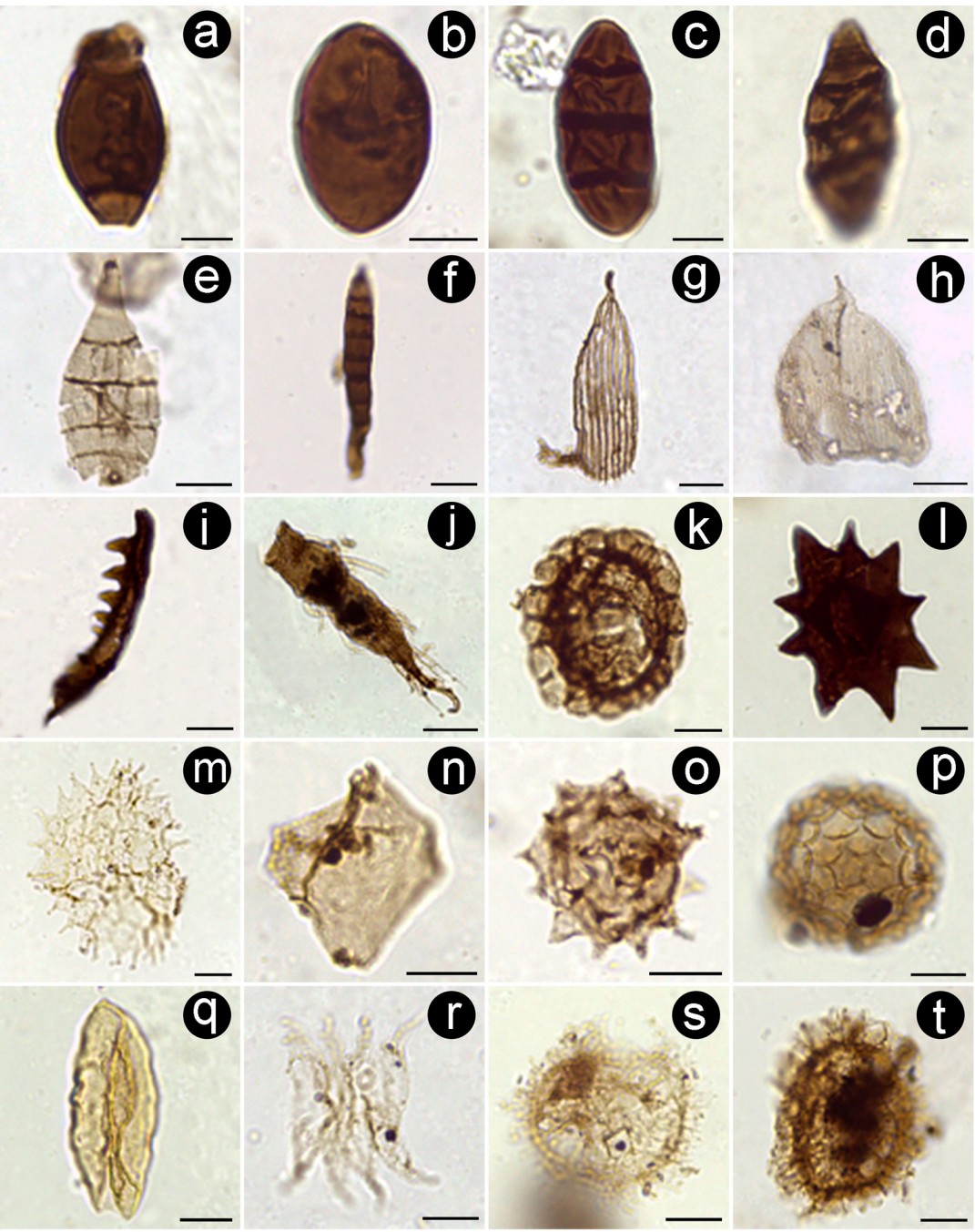

**Fig 7. Microscope images of selected non-pollen palynomorphs from Casa Grande Formation.** (A) *Biporipsilonites krempii* (PAL JUA (P) Cz/Plg. N° 001b(Eo)+10 μm: 2–142.5). (B) *Inapertisporitese edigeri* (PAL JUA (P) Cz/Plg. N° 001b(Eo)+10 μm: 6–145.5). (C) *Multicellites crassisporus* (PAL JUA (P) Cz/Plg. N° 001d(Eo)+10 μm: 20.8–129.2). (D) *Quilonia* cf. *Q. allepeyensi* (PAL JUA (P) Cz/Plg. N° 001b(Eo)+10 μm: 4.1–126.7). (E) *Polycellaesporonites bellus* (PAL JUA (P) Cz/Plg. N° 001e(Eo) +10 μm: 7–141.5). (F) *Scolecosporites modicus* (PAL JUA (P) Cz/Plg. N° 001b(Eo)+10 μm: 7.5–153.4). (G) Scales of Lepidopteran wings (PAL JUA (P) Cz/Plg. N° 001e(Eo)+25 μm: 11–143.9). (H) Scales of Lepidopteran wings (PAL JUA (P) Cz/ Plg. N° 001d(Eo)+25 μm: 19.8–141.5). (I) Scolecodont-like palynomorph (PAL JUA (P) Cz/Plg. N° 001e(Eo)+25 μm: 2–153.9). (J) *Hydrozetes* adult leg (PAL JUA (P) Cz/Plg. N° 001d(Eo)+25 μm: 16.5–144.5). (K) *Filinia*-type resting eggs (PAL JUA (P) Cz/ Plg. N° 001d(Eo)+25 μm: 23.2–155). (L) Compact type sclerocyte (PAL JUA (P) Cz/Plg. N° 001d(Eo)+25 μm: 22.5–152). (M) *Pediastrum boryanum* (PAL JUA (P) Cz/Plg. N° 001d(Eo)+25 μm: 4–149.5). (N) *Mougeotia* sp. (PAL JUA (P) Cz/Plg. N° 001a (Eo)+10 μm: 6–145). (O) *Planctonites stellarius* (PAL JUA (P) Cz/Plg. N° 001d(Eo)+10 μm: 18–145). (P) *Stigmozygodites ministigmosus* (PAL JUA (P) Cz/Plg. N° 001d(Eo)+25 μm: 8.8–138). (Q) *Spirogyra* sp. 6 Martínez et al. 2008 (PAL JUA (P) Cz/ Plg. N° 001d(Eo)+10 μm: 23.9–134.9). (R) *Scenedesmus* sp. (PAL JUA (P) Cz/Plg. N° 001e(Eo)+25 μm: 3.5–147.5). (S)

Indeterminate dinoflagellate cyst 1 (PAL JUA (P) Cz/Plg. N° 001d(Eo)+25 µm: 21.7–141.8). (T) Indeterminate dinoflagellate cyst 2 (PAL JUA (P) Cz/Plg. N° 001c(Eo)+25 µm: 15–146.6). Scale bars: 10 µm (except A and H: 5 µm and 25 µm, respectively). Each morphotype name is followed by slide number and coordinates of Leica DM500 microscope held at the Museo Argentino de Ciencias Naturales, Buenos Aires, Argentina.

slightly younger than mid Eocene, yet further analysis (either biostratigraphic or radiometric) is needed. However, the presence of typical species of the Paleocene-Eocene (e.g., *Verrustephanoporites simplex*) along with the complete absence of key open-habitat representatives from the Neogene such as the widely distributed spiny pollen grains of the Asteroideae (daisy family) gives us the notion that the Casa Grande Fm. might not be younger than Paleogene.

Overall, we assume that our reconstructed mid Eocene plant community has no close analogue in the modern South American vegetation, although a broad resemblance to the seasonally-dry forest of the Chaco becomes increasingly probable. This is largely because of the co-occurrence of Fabaceae, Malvaceae Bombacoideae and Ulmaceae *Phyllostylon*, which grow today particularly across the humid and dry Chaco. Their occurrence, along with the presence of several species of palms (Arecaceae), lead us to suggest subtropical or tropical conditions and frost-free winters. More importantly, our analysis supports previous assumptions

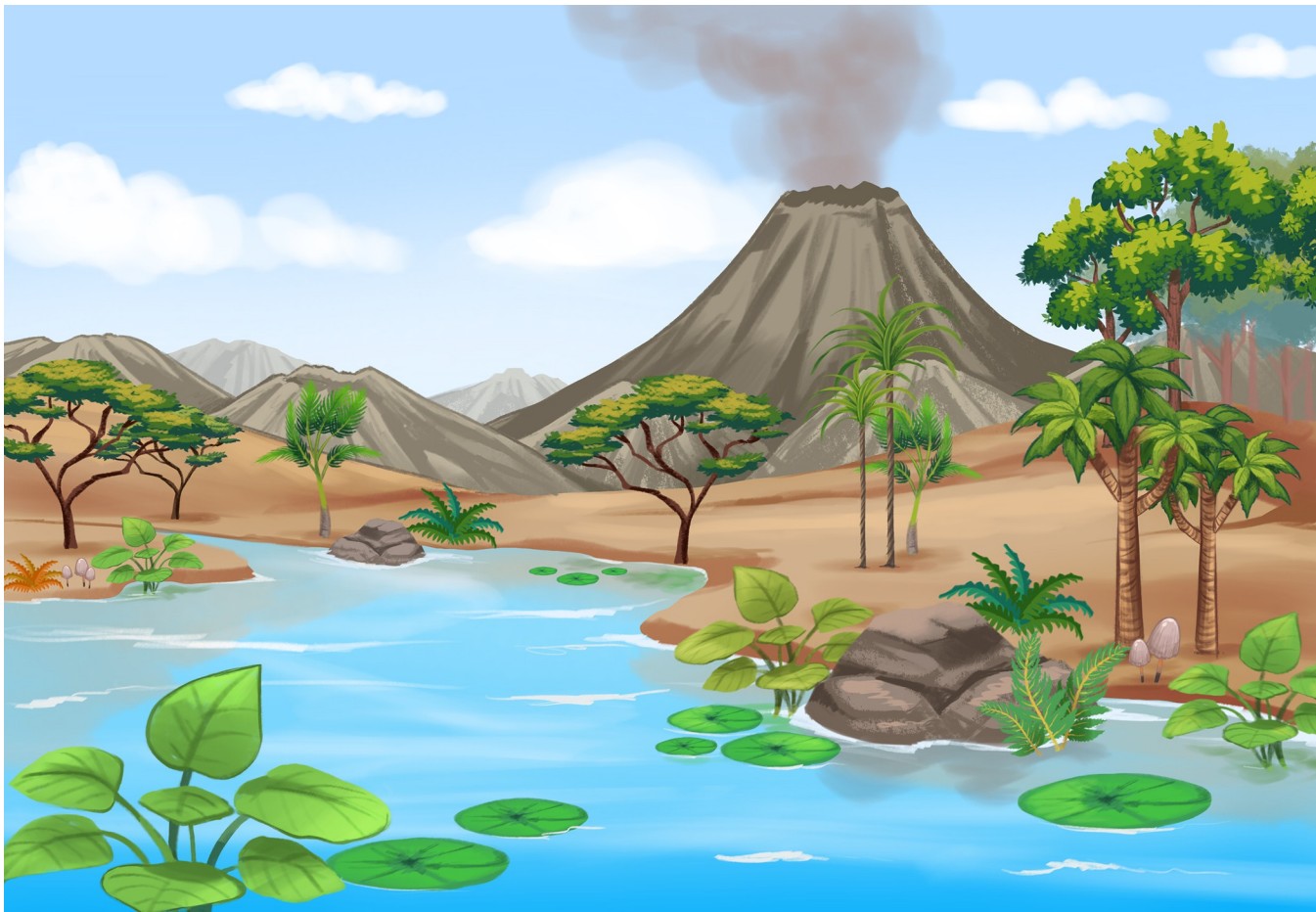

**Fig 8. Reconstruction of southern Central Andean landscapes during the Eocene.** The nearest living representatives of the discovered fossils indicate the presence of trees and shrubs nearby a vegetated pond with ferns and floating plants.

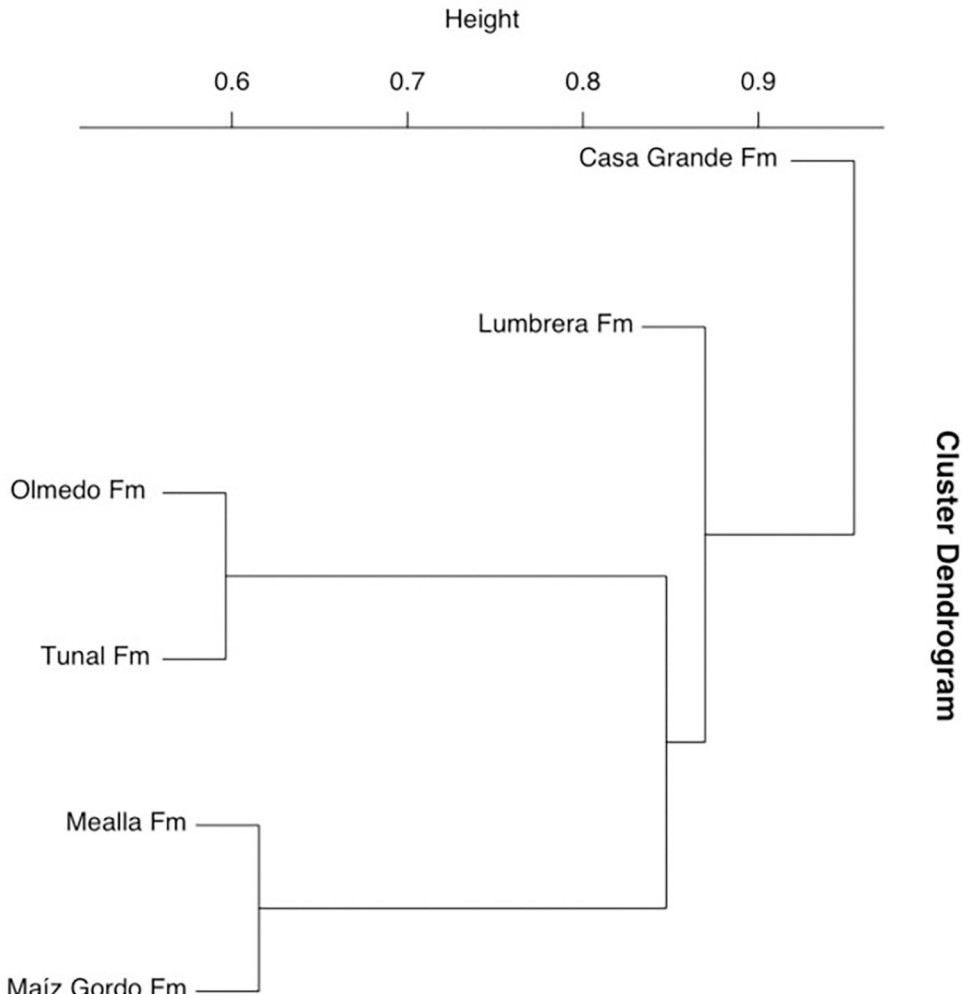

**Fig 9. Cluster dendrogram.** Hierarchical clustering showing the similarities among the spore-pollen records from the Casa Grande 1 sequence (this study) and those previously published from other nearby Paleogene sites based on presence/absence data (S2 Table). The number of samples and morphotypes of each site is given in Materials and Methods. The heterogeneity of either the fossil record and the sampling effort may influence the outcome of the hierarchical clustering, hence results should be interpreted accordingly.

indicating that the threshold elevation at which the orographic rain shadow effect became established at these latitudes, post-date the accumulation of the Casa Grande 1 Formation.

## Supporting information

**S1 Appendix. Systematic section.** This section includes brief and informal descriptive remarks and dimensions on selected species recorded in the Casa Grande 1 assemblage. (DOCX)

**S1 Table. Identified mophotype list and frequencies (%).** Frequencies are based on a count of 546 palynomorphs. A black circle indicates frequencies lower than 0.1%. (DOCX)

**S2 Table. Data matrix (presence/absence) of pollen and spores from the Paleogene of the Salta Group.** OF: Olmedo Formation, TF: Tunal Formation, MF: Mealla Formation, MGF:

Maíz Gordo Formation, LF: Lumbrera Formation, CGF: Casa Grande Formation (this work). (DOCX)

**S3 Table. Comparative sampling-count chart among Grupo Salta Basin sedimentary units.** (DOCX)

## Acknowledgments

We thank S. Mirabelli for processing palynological samples, Luz M. Tapia for drawing the Eocene reconstruction and the Editor and two anonymous reviewers for their helpful comments. We also thank Mirta Quattrocchio, Eduardo Bellosi and Paula Narvaez for providing us hard-to-find papers.

## Author Contributions

**Conceptualization:** Lilia R. Mautino.

**Investigation:** Mariano J. Tapia, Ezequiel E. Farrell, Lilia R. Mautino, Cecilia del Papa, Viviana D. Barreda, Luis Palazzesi.

**Supervision:** Cecilia del Papa, Luis Palazzesi.

**Validation:** Mariano J. Tapia, Ezequiel E. Farrell, Viviana D. Barreda.

**Writing – review & editing:** Luis Palazzesi.

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
