## [Decision Letter · Decision Letter 0]

12 Dec 2022

PONE-D-22-29456A snapshot of mid Eocene landscapes in the southern Central Andes: spore-pollen records from Casa Grande Formation (Jujuy, Argentina)PLOS ONE

Dear Dr. Palazzesi,

Thank you for submitting your manuscript to PLOS ONE. After careful consideration, we feel that it has merit but does not fully meet PLOS ONE’s publication criteria as it currently stands. Therefore, we invite you to submit a revised version of the manuscript that addresses the points raised during the review process.

We look forward to receiving your revised manuscript.

Kind regards,

Gongle Shi, Ph.D.

Academic Editor

PLOS ONE

Journal Requirements:

4. We note that Figures 1 and 2 in your submission contain [map/satellite] images which may be copyrighted. All PLOS content is published under the Creative Commons Attribution License (CC BY 4.0), which means that the manuscript, images, and Supporting Information files will be freely available online, and any third party is permitted to access, download, copy, distribute, and use these materials in any way, even commercially, with proper attribution. For these reasons, we cannot publish previously copyrighted maps or satellite images created using proprietary data, such as Google software (Google Maps, Street View, and Earth). For more information, see our copyright guidelines: http://journals.plos.org/plosone/s/licenses-and-copyright.

a. You may seek permission from the original copyright holder of Figures 1 and 2 to publish the content specifically under the CC BY 4.0 license.  

Reviewers' comments:

Reviewer's Responses to Questions

**Comments to the Author**

1. Is the manuscript technically sound, and do the data support the conclusions?

Reviewer #1: Partly

Reviewer #2: Yes

2. Has the statistical analysis been performed appropriately and rigorously? 

Reviewer #1: Yes

Reviewer #2: Yes

3. Have the authors made all data underlying the findings in their manuscript fully available?

Reviewer #1: Yes

Reviewer #2: Yes

4. Is the manuscript presented in an intelligible fashion and written in standard English?

Reviewer #1: Yes

Reviewer #2: Yes

5. Review Comments to the Author

Reviewer #1: Farrell et al. documented the mid Eocene floristic composition of the Casa Grande Formation in the Puna region for the first time, and reconstructed its landscape. The authors found about 70 palynomorphs including a few Gondwanan taxa, forming a frost-free and humid to seasonally dry ecosystem near a lacustrine environment. This is a preliminary study with only one analyzable pollen sample among seven collected samples, and there has some space for improvement.

1. The hierarchical clustering cannot be used for comparison in this study, because this research has only one sample, while others have some samples (e.g., In ref. 16, it has five samples. In ref. 17, it has four samples? In ref. 19, it has 16 (of 26 samples) yielding palynomorphs. In ref. 20, it has 5 (of 9 samples) yielding palynomorphs). This is sampling bias which the authors did not discuss in the text. The authors will have to revise/delete corresponding text.

2. As the Casa Grande assemblage is quite different from those produced from older sediments, it is useful for the authors to add a systematics section to describe some important taxa.

3. For the reconstructed landscape, the authors can draw a landscape model for a better visualization, and maybe also for the five compared localities.

4. For the other comments, please refer to the annotated PDF.

Reviewer #2: The hierarchical clustering is lack of percentage values, instead, only depends on the absence/presence data, that seems not convictive. I suggest publication after providing the pollen counting data and hope my comments are helpful.

Other minor comments please refer to the attached comments in the pdf.

6. PLOS authors have the option to publish the peer review history of their article (what does this mean?). If published, this will include your full peer review and any attached files.

Reviewer #1: No

Reviewer #2: No

---

## [Author Response · Author response to Decision Letter 0]

14 Feb 2023

Dear Dr. Gongle Shi,

We have revised our manuscript following your and the reviewers' suggestions in red.

Please, include in our “Financial disclosure” the following sentence: “This study was partially supported by CONICET (PIP 666-CONICET grant to C. del Papa) and by CONICET-GFZ/DFG StRaTEGy (Surface, processes, tectonics, and georesources: The Andean foreland basin of Argentina).”

Response #1: We have checked that our manuscript meets PLOS ONE’s style requirements.

Response #2: We have now included the permits we obtained to access the samples in our Methods section..

Response #3: The “Data Availability statement” should include “All relevant data are within the manuscript and Supplementary Table 1”.

4. We note that Figures 1 and 2 in your submission contain [map/satellite] images which may be copyrighted. All PLOS content is published under the Creative Commons Attribution License (CC BY 4.0), which means that the manuscript, images, and Supporting Information files will be freely available online, and any third party is permitted to access, download, copy, distribute, and use these materials in any way, even commercially, with proper attribution. For these reasons, we cannot publish previously copyrighted maps or satellite images created using proprietary data, such as Google software (Google Maps, Street View, and Earth). For more information, see our copyright guidelines: http://journals.plos.org/plosone/s/licenses-and-copyright.

Response #4: We removed Fig 1, which was the only copyrighted figure, and replaced it with another taken from USGS National Map Viewer (public domain). Figure 2 was taken by one of us (C. del Papa) and does not need copyright permission as it has never been published before. Figure 8 was downloaded from Vecteezy.com (copyrighted free, and only attribution is required).

Thanks for editing our manuscript,

Kind regards,

Luis Palazzesi (on behalf of all authors)

Reviewer #1: Farrell et al. documented the mid-Eocene floristic composition of the Casa Grande Formation in the Puna region for the first time, and reconstructed its landscape. The authors found about 70 palynomorphs including a few Gondwanan taxa, forming a frost-free and humid to seasonally dry ecosystem near a lacustrine environment. This is a preliminary study with only one analyzable pollen sample among seven collected samples, and there has some space for improvement.

1. The hierarchical clustering cannot be used for comparison in this study, because this research has only one sample, while others have some samples (e.g., In ref. 16, it has five samples. In ref. 17, it has four samples? In ref. 19, it has 16 (of 26 samples) yielding palynomorphs. In ref. 20, it has 5 (of 9 samples) yielding palynomorphs). This is sampling bias which the authors did not discuss in the text. The authors will have to revise/delete corresponding text.

Response #1: We appreciate Reviewer #1 comments. In the new version, we have discussed in the main manuscript (lines 146-148) the sampling bias that our analysis faced. We are aware that the comparison of sedimentary sections with different sampling efforts may cause misleading results. However, the number of palynomorphs recovered, which is the main source of comparisons, is relatively similar among sections. We have now included the number of samples and the number of palynomorphs used for the hierarchical clustering (lines 149-151, S3 Table). And we have now explicitly expressed that results should be interpreted with caution.

2. As the Casa Grande assemblage is quite different from those produced from older sediments, it is useful for the authors to add a systematics section to describe some important taxa.

Response #2: The new version of our manuscript now includes a systematic section (Supplement 1 Appendix) to describe and/or comment on some key species.

3. For the reconstructed landscape, the authors can draw a landscape model for a better visualization, and maybe also for the five compared localities.

Response #3: We have now included a reconstructed landscape based on the interpretations of the botanical affinity of the discovered palynomorphs from the Casa Grande Formation (Fig. 8). The reconstruction of landscapes from other Paleogene sites may be an interesting feature to include, although it goes beyond the main goal of our work. 

4. For the other comments, please refer to the annotated PDF.

Response #4: We included all modifications suggested by Reviewer #1 in this new version of the manuscript. We appreciate Reviewer #1 corrections and suggestions; please, thank him/her on our behalf.

Reviewer #2: The hierarchical clustering is lack of percentage values, instead, only depends on the absence/presence data, that seems not convictive. I suggest publication after providing the pollen counting data and hope my comments are helpful.

Other minor comments please refer to the attached comments in the pdf.

Response #5: We have now provided a pollen-counting dataset (S2 Table). We used presence/absence data for our hierarchical clustering analysis because not all publications provided pollen-counting data, but a list of species, instead. We have now specified this in the methods, and commented on the limitations our approach faced. 

We have also included a new (S3 Table) with the sedimentary units used for comparison along with the number of samples and species that each of them includes. We appreciate all the comments of Reviewer #2 from the pdf, which improved the quality of our work. Please, thank him/her on our behalf.

---

## [Editor Report · Decision Letter 1]

2 Mar 2023

A snapshot of mid Eocene landscapes in the southern Central Andes: spore-pollen records from the Casa Grande Formation (Jujuy, Argentina)

PONE-D-22-29456R1

Dear Luis,

We’re pleased to inform you that your manuscript has been judged scientifically suitable for publication and will be formally accepted for publication once it meets all outstanding technical requirements.

Kind regards,

Gongle Shi, Ph.D.

Academic Editor

PLOS ONE
---

## [Editor Report · Acceptance letter]

12 Mar 2023

PONE-D-22-29456R1 

A snapshot of mid Eocene landscapes in the southern Central Andes: spore-pollen records from the Casa Grande Formation (Jujuy, Argentina) 

Dear Dr. Palazzesi:

I'm pleased to inform you that your manuscript has been deemed suitable for publication in PLOS ONE. Congratulations! Your manuscript is now with our production department. 

Kind regards, 

on behalf of

Dr. Gongle Shi 

Academic Editor

PLOS ONE